# Fast Response Isopropanol Sensing Properties with Sintered BiFeO_3_ Nanocrystals

**DOI:** 10.3390/ma13173829

**Published:** 2020-08-30

**Authors:** Hongxiang Xu, Junhua Xu, Junlin Wei, Yamei Zhang

**Affiliations:** School of Materials Science and Engineering, Jiangsu University of Science and Technology, Zhenjiang 212003, China; hongxiangxu@just.edu.cn (H.X.); 182060012@stu.just.edu.cn (J.W.)

**Keywords:** bismuth ferrite, gas sensor, nanocrystals, isopropanol

## Abstract

BiFeO_3_ nanocrystals were applied as the sensing material to isopropanol. The isopropanol sensor based on BiFeO_3_ nanocrystals shows excellent gas-sensing properties at the optimum working temperature of 240 °C. The sensitivity of as-prepared sensor to 100 ppm isopropanol is 31 and its response and recovery time is as fast as 6 and 17 s. The logarithmic curves of the sensitivity and concentration of BiFeO_3_ sensors are a very good linear in the low detection range of 2–100 ppm. In addition, the gas sensing mechanism is also discussed. The results suggest that the BiFeO_3_ nanomaterial can be potentially applied in isopropanol gas detection.

## 1. Introduction

With the fast economic growth accompanied by the large demands of industry activities, the emission of harmful substances are increasing, resulting in serious environment problems, such as air pollution. Volatile organic compounds (VOCs) are important atmospheric pollutants ubiquitous around us due to their vast application in production and living activities of construction, transportation, furnishing, etc., and can cause human-health problems. Isopropanol, added by IARC (part of WHO) into the lists of group 1 and group 3 carcinogens, is a member of the widely used VOC-family [1]. It is very harmful for a human to be exposed to isopropanol. A low concentration (below 400 ppm) will stimulate the upper respiratory tract and cause eye discomfort, while a high concentration will suppress the central nervous system and cause severe vomiting, respiratory decline, and internal bleeding [2]. Therefore, as an effective approach, using gas sensors to realize the precise detection and early warning of isopropanol is of great importance.

During the past years, scientists have done a lot of researches on high-sensitive gas sensors [3,4], especially miniaturized smart sensors with fast response and real-time monitoring characteristics [5]. Gas-sensitive materials are the basic core components of gas sensors. After decades of research and development, binary metal oxide semiconductor materials have taken center stage, such as SnO_2_ [6], ZnO [7], TiO_2_ [8], and Fe_2_O_3_ [9], which show many advantages of high sensitivity, simple design, low cost, efficiency, compatibility, and portability. However, applications have seen the unsatisfactory working temperature, long-term stability, and working life of binary metal oxides as above [10]. To improve these weaknesses, several types of architectures such as mesoporous ball-flower structures [11] and hollow nanospheres [12] have been designed. Although some gas sensitive parameters such as gas sensitivity can be optimized, other important features are likely to be compromised. Therefore, instead of binary oxides, seeking other materials inherently with excellent gas sensing properties is another important way to improve gas sensors. 

Since perovskite-structure oxides have excellent semiconductor properties, high temperature stability and rapid oxygen mobility [13,14,15,16], they are ideal candidates. BiFeO_3_ is a distorted perovskite oxide which possesses both ferroelectricity (T_C_ = 1103 K) and G-type antiferromagnetism (T_N_ = 643 K) [17]. BiFeO_3_ is also a p-type semiconductor and its optical band gap is approximately 2 and 3 eV for nanoparticles and single crystals, respectively [18,19]. Given the oxides’ commonalities, lots of oxygen vacancies are in the body and the surface of BiFeO_3_, while the surface oxygen vacancies provide reliable locations for external oxygen molecules. Meanwhile, BiFeO_3_ is a kind of oxide material with a fast oxygen vacancy diffusion rate [20,21]. Therefore, BiFeO_3_ exhibits excellent gas sensing properties. However, only a handful of relevant articles have been published, most of whose application objectives are inorganic gases and limited VOCs such as acetone and ethanol [22,23,24,25,26,27,28,29,30]. Most of these reports indicate that BiFeO_3_ is very sensitive to alcohols and ketones and the gas sensitivities of BiFeO_3_ sensors are influenced by their morphology and particle size. Tong Tong [27] found that the sensitivities of BiFeO_3_ gas sensors to 50 ppm ethanol based on their disk-shaped particles (2 μm) by the hydrothermal method and nanoparticles (100 nm) by the co-precipitation method are respectively 3.3 and 4 at 260 °C. Yu Xuelian [29] controlled the size of nanoparticles by adjusting the growth time using the sol-gel method to construct BiFeO_3_ tubular sensors and found that the smaller the particle size of BiFeO_3_ is, the greater the sensitivity is. The sensitivity of 30 nm (100 nm) particles to 50 ppm ethanol and acetone at 260 °C are 40 (27) and 36 (24), respectively. It can be seen that the growth time and temperature need to be reduced in order to reduce the particle size and improve the sensitivity. In those reported BiFeO_3_ ethanol sensors, BiFeO_3_ nanoparticles have better gas sensing property than other nanostructures. It is probably because smaller nanoparticles have a larger specific surface area and abundant oxygen vacancy defects. So far, the BiFeO_3_ gas sensitive material is limited to only a few organic gases and is still worth further exploring for many organic gases. Considering the gas sensing advantages of BiFeO_3_ nanoparticles and the toxicity of isopropanol, BiFeO_3_ nanocrystals are applied as the sensing material to isopropanol in this paper. Different gas sensitive parameters such as sensitivity and response/recovery time are studied, and the gas sensing mechanism is also discussed by surface chemical reaction and gas adsorption/desorption between the external gas and the surface of BiFeO_3_ nanocrystals.

## 2. Materials and Methods 

### 2.1. Preparation of BiFeO_3_ Nanocrystals

BiFeO_3_ nanocrystals were synthesized by a sol-gel method. All chemicals used were of analytical grade. First, 0.006 mol Bi(NO_3_)_3_·5H_2_O (2.938 g) was added into the dilute nitric acid (2 ml concentrated nitric acid whose content is 65–68% was mixed by 8 mL of water) and stirred for half an hour to obtain a colorless transparent solution (Bi-based solution). Next, 0.006 mol Fe(NO_3_)_3_·9H_2_O (2.461 g) was dissolved in deionized water (10 mL) to a form a transparent red solution (Fe-based solution). Then, the Bi-based solution and the Fe-based solution were mixed together to form a transparent yellow solution. A few drops of 3% polyvinyl alcohol solution and tartaric acid were successively added into the transparent yellow solution as the surfactant and the complexing agent, respectively. Here, the molar amount of tartaric acid was greater than the total molar number of metal ions of nitrates. The mixed solution was stirred for about 10 h at room temperature and then heated at 90 °C until the sol changed into a bright shining yellow dry gel. The dry gel was ground in an agate mortar for 1 h and then sintered for 4 h at different temperatures (500 and 550 °C).

### 2.2. Characterization

Thermogravimetric analysis and differential thermal analysis (TG-DTA) were performed from 25 to 800 °C in air to determine the thermal behavior of the dry gel. The surface morphologies of BiFeO_3_ nanocrystals were observed by a field emission scanning electron microscopy (FESEM, Merlin Compact, Carl Zeiss, Oberkochen, Germany. An XRD-6000 (Shimadzu Corporation, Tokyo, Japan) X-ray diffractometer (Cu Kα, λ = 1.5406 Å, scanning range 20–70°, 4°/min) was applied to analyze the crystallinity and phase purity of the synthesized bismuth ferrite. An X-ray photoelectron spectrum (XPS) was used to characterize the surface compositions and corresponding element valence of BiFeO_3_ nanocrystals. 

### 2.3. Fabrication and Measurement of Gas Sensors

Two hundred mg BiFeO_3_ nanocrystals were mixed with deionized water (0.2 mL) by grinding in an agate mortar to form a paste. The mixed paste was brushed thinly on the surface of the Ag-Pd interdigital electrode on the Al_2_O_3_ substrate. After drying, the planar BiFeO_3_ gas sensors were obtained. 

Before measurement, the dried BiFeO_3_ sensors were aged at 200 °C for 10 h in air. The gas sensitive parameters (sensitivity, response time, recovery time, working temperature, stability, and selectivity) of the BiFeO_3_ sensors were obtained on a CGS-1 TP intelligent gas sensing analysis system (Beijing Elite Tech Co., Ltd., Beijing, China). Here, the R_g_/R_a_ ratio was used to evaluate the sensitivity, where R_g_ and R_a_ were the sensor resistance in the measured gas and in fresh air, respectively. The response or recovery time was the time when the resistance change of BiFeO_3_ sensor reached 90% of (R_g_ − R_a_) [31]. For more details about the measurement of the sensors, please refer to our previous work [26].

## 3. Results and Discussion

### 3.1. Growth Analysis 

TG and DTA curves of the dry gel are shown in Figure 1. When the temperature increases from room temperature to 100 °C, the weight hardly changes. The endothermic peak at 144 °C is attributed to the evaporation of a small amount of water and acid attached to the gel, and the relevant weight loss ratio is about 10%. The exothermic peak at 210 °C on the DTA curve corresponds to the collapse of gel networks and the combustion of most organic materials producing a large number of nitrogen and carbon oxides [32]. When the temperature reaches 300 ℃, the weight loss ratio of the dry gel is 45%. A tiny weight loss (7%) between 300 and 500 °C means the release of carbon oxides, indicating that there are still a small number of residual substances which has not been completely decomposed. No more weight loss is observed when the temperature is above 500 °C. The exothermic peak at 502 °C possibly indicates the full formation of the crystal. Therefore, to control the grain size, different grown temperatures (500 and 550 °C) are adopted as the sintering temperature of BiFeO_3_ nanocrystals.

### 3.2. Characterization 

Figure 2 shows the surface micrographs of as-prepared BiFeO_3_ nanocrystals sintered at 500 and 550 °C. It is obvious that BiFeO_3_ nanocrystals have irregular surface morphologies and their average size is approximately 100 nm. The nanocrystals appear to stick to each other and agglomerate in a large number. In addition, the nanocrystals grown at 550 °C are a little bigger and less agglomerated than those grown at 500 °C. However, in general, the morphologies of nanocrystals at the two temperatures are similar without much distinction.

X-ray diffraction patterns of BiFeO_3_ nanocrystals sintered at 500 and 550 °C are shown in Figure 3. It is obvious that all the reflection peaks of BiFeO_3_ nanocrystal powders match with the spectrum of JCPDS Card No. 86-1518, and all diffraction peaks of BiFeO_3_ nanocrystals are obviously shifted to the right by about 0.13°, suggesting that BiFeO_3_ nanocrystal powders are of the rhombohedral distorted perovskite structure (Space group R3c). It is probably because there are a large number of oxygen vacancies in the body and on the surface of BiFeO_3_ nanocrystals greatly increasing the lattice distortion and leading to the diffraction peak shift to the right. The diffraction peaks of two sintered powders did not deviate significantly and the consistency of their peak heights and peak widths once again indicates the comparability of the two nanocrystals.

Given the consistency of the 500 and 550 °C nanocrystals, here we choose the former for the XPS study. Figure 4a shows the XPS spectrum of Bi, Fe, O, and C, among which C 1 s is from the measuring environment. For Bi, Fe, and O elements, Gaussian deconvolution fitting is used to decompose two peaks. Figure 4b shows two photoemission peaks of Bi 4f at 159.1 and 164.5 eV. Therefore, the spin orbit splitting energy is 5.4 eV in the core level spectra between Bi 4f_7/2_ and Bi 4f_5/2_, which indicates that Bi is +3 [33]. Figure 4c shows the spectrum of Fe 2p with two peaks of Fe 2p_3/2_ and Fe 2p_1/2_ at 709.6 and 724.2 eV, suggesting the presence of Fe^2+^ cations. In addition, a binding energy is also fitted at 712.0 eV, corresponding to Fe^3+^ cations. Therefore, Fe^2+^ and Fe^3+^ cations coexist in BiFeO_3_ nanocrystals and oxygen vacancies appear to compensate the electrical neutrality. Figure 4d shows the O 1 s XPS spectrum. The two peaks at 529.7 and 531.8 eV are derived from lattice oxygen and chemisorbed oxygen, respectively [34]. Thus, the XPS results demonstrate the presence of Bi, Fe, O in the form of Bi^3+^, Fe^3+^, Fe^2+^, oxygen ions and O vacancies, respectively.

### 3.3. Gas Sensing Performance

BiFeO_3_ sensors express the electrical characteristics of a typical semiconductor that the resistance value of sensors in air decreases with the temperature increasing. The charge carriers of BiFeO_3_ are holes produced by Bi^3+^ cation vacancy defects $\left(V_{\mathrm{Bi}}^{x}\right)$
$\left(V_{\mathrm{Bi}}^{x}\to V_{\mathrm{Bi}}^{\prime \prime \prime }{+3h}^{.}\right)$(Kroger-Vink defect notation) [35]. Since bismuth ferrite is a non-stoichiometric oxide, oxygen vacancy defects have been proven to exist on its surface and in its body [21]. At different temperatures, the number of oxygen ions adsorbed on the surface of BiFeO_3_ nanocrystals is different, so the number of oxygen ions reacting with the isopropanol gas molecules is different too. The isopropanol sensing performance of BiFeO_3_ gas sensors were investigated at different operating temperatures to confirm the optimum condition. Figure 5a shows the gas response (assessed by sensitivity) of BiFeO_3_ gas sensors to 50 ppm isopropanol from 200 to 280 °C. With the temperature increasing, the isopropanol sensitivities of two BiFeO_3_ sensors are both found to increase at first, undergo the maximum value at 240 °C, and then decrease gradually. Since isopropanol gas molecules need a certain amount of thermal activation energy to react with the surface absorbed oxygen species. When the operating temperature is too low, insufficient thermal energy and low amount of absorbed oxygen ions on the sensor surface will inhibit the reaction between isopropanol gas molecules and surface adsorbed oxygen ions [36]. However, when the operating temperature is too high, the desorption rate of oxygen gas is enhanced. As a result, some adsorbed oxygen species may escape before reacting with isopropanol gas molecules so that the response to isopropanol decreases correspondingly [37]. Therefore, the optimal operating temperature is at an equilibrium point between the adsorption and desorption process. The largest response values for two sensors based on BiFeO_3_ nanocrystals sintered at 500 and 550 °C both appear at 240 °C. Therefore, 240 °C is considered as the optimum working temperature. Figure 5b shows the typical dynamic resistance-time gas response curves of two sensors at 240 °C to different concentrations of isopropanol (2–300 ppm). The initial resistances of BiFeO_3_ sensors grown at 500 and 550 °C in air are 2.2515 × 10^6^ and 3.5876 × 10^6^ Ω, respectively. The sensitivity-time curve is automatically obtained by the gas sensitive testing software according to the formula S = R_g_/R_a_. The sensitivity of two sensors is almost the same after calculation by the testing software and increase gradually with the isopropanol concentration increasing in the range of 2–100 ppm. The response value corresponds approximately to 4.8, 10, 14, 18, 26, 30, 31 for the isopropanol concentration of 2, 5, 10, 20, 50, 80, 100 ppm. The sensitivity of BiFeO_3_ gas sensors to the lowest testing concentration (2 ppm) of isopropanol reaches 4.8 which can be obviously observed, meaning that the detection limit of BiFeO_3_ gas sensor is as low as 1 ppm level in our experiment error. When the concentration is greater than 200 ppm, the sensitivity no longer increases. This indicates that oxygen ions adsorbed on the surface of BiFeO_3_ nanocrystals have reached its saturation point, implying that bismuth ferrite is very sensitive to isopropanol at a low concentration. During the exchanging cycle between isopropanol and ambient air, the resistance almost restores its initial value immediately after isopropanol is released, which suggests the good stability of BiFeO_3_ sensors. 

In order to further discuss the isopropanol response of BiFeO_3_ sensors under different concentrations, the traditional empirical formula of oxide semiconductor gas sensor is applied to fit the sensitivity-concentration curves. In general, the sensitivity-concentration dependence of metal oxide semiconductor is empirically represented as: (1)$$S\;=\;a{\left(C\right)}^{b}\;+\;1$$ where S represents the sensitivity, C is the concentration of isopropanol, and the ideal constant value (a) denotes a prefactor depending on the gas sensing material [8,11,38]. At the optimum working temperature of 240 °C, the equation can be rewritten in the logarithm function form: (2)log(S−1)= b log C + log a

Based on Figure 5b, we obtained the sensitivity-time curve at different concentrations. The above formula is used to fit the relation between sensitivity and concentration. Figure 5c reveals that log(S−1)−logC curves of two BiFeO_3_ sensors are linear in the concentration range of 2–100 ppm. The b values of two sensors are respectively 0.38 and 0.43, approaching to 0.5, suggesting that oxygen ions on the surface of BiFeO_3_ nanocrystals are mainly composed of O^2–^. The measured correlation coefficients R^2^ of two gas sensors are respectively 0.986 and 0.996, indicating good linear relations. 

The R_g_/R_a_ ratio is greater than 1 for the p-type semiconductor in reducing gas, and the resistance of the gas sensor is kept relatively steady in air before isopropanol gas molecules are injected. Upon the injection of isopropanol gas molecules, the sensor resistance increases fast. When the resistance of BiFeO_3_ sensor reaches the maximum value, it will maintain a relatively stable value. When isopropanol gas molecules are removed, the resistance of the sensor decreases rapidly. Therefore, the sensor has a very short response and recovery time. Figure 5d shows that the response and recovery time of two BiFeO_3_ sensors toward 100 ppm isopropanol at 240 °C are 6 and 17 s. These results show that BiFeO_3_ nanocrystals can respond quickly to isopropanol in a few seconds and are very suitable for making fast isopropanol gas sensors.

In practical application, the reversibility and long-term stability are also important properties that determine the practicability and cost performance of sensors [1,2,12,13,38]. Thus, the six-cycle experiment of BiFeO_3_ sensors to 100 ppm isopropanol at 240 °C was carried out, as shown in Figure 6a. The curves show that the sensitivity almost has no change after six exchanging cycles between air and isopropanol. The sensitivity in the end of each cycle can recover to the initial value, indicating that the sensor has a good reversibility. Figure 6b shows that the sensitivity almost remains constant during the test days, revealing good long-term stabilities of both sensors to 100 ppm isopropanol at 240 °C during one month.

The selectivity to a target gas among various gases is a crucial parameter to evaluate the gas sensing performance. The selectivity of BiFeO_3_ sensors was investigated by the response to VOCs including isopropanol, methanol, formaldehyde, cholamine, acetic acid, ammonia, dimethylformamide (DMF), and acrylic acid. As shown in Figure 7, the maximum response value of BiFeO_3_ sensor sintered at 500 °C is respectively 26, 12.2, 10.2, 7.2, 6.2, 3.8, 5.6, 8.7 to isopropanol, methanol, formaldehyde, cholamine, acetic acid, ammonia, DMF, and acrylic acid at 240 °C. In addition, the 550 °C sintered BiFeO_3_ shows the same trend. It shows that the response value to isopropanol is prominently higher than that to other gases. This phenomenon might be attributed to the intrinsic nature of gas species such as the molecular weight, molecular structure, and bond strength [36]. Since the bond strength of C=O (799 kJ/mol) is much higher than other bonds, e.g., C–O, C–H, C–C, H–O, C–N, C=C, N–H corresponding to 358, 411, 346, 459, 305, 602, 386 kJ/mol, respectively, it is more difficult for organic gases containing C=O such as formaldehyde, acetic acid, DMF, and acrylic acid to be decomposed than other organic gases such as isopropanol and methanol. For isopropanol and methanol, the former with larger molecules can be more easily adsorbed to react with oxygen ions. Therefore, the sensors show higher response to isopropanol than to methanol. 

At present, the understanding of the gas sensing mechanism of single phase oxide semiconductor such as BiFeO_3_ is based on the electrical resistance change during adsorption and desorption in different gas atmospheres. When the BiFeO_3_ sensor is exposed to air, oxygen molecules will be adsorbed on the surface of BiFeO_3_ nanocrystals and they can be ionized into $O_{2}^{-}$, $O^{-}$, $O^{2-}$ by capturing free electrons of the conduction band. These oxygen species ($O_{2}^{-}$,$\;O^{-}$,${\;O}^{2-}$) are formed on the surface of BiFeO_3_ nanocrystals, resulting in the reducing of electron concentration and the development of the electron depletion layer between air and the surface of BiFeO_3_ sensor. It can be described as follows [3,6,9,13,27,28]: (3)$$O_{2}\left(\mathrm{gas}\right)\to O_{2}(\mathrm{ads}$$
(4)$$O_{2}\left(a\mathrm{ds}\right)+e^{-}\to O_{2}^{-}\left(\mathrm{ads}\right)$$
(5)$$O_{2}^{-}\left(\mathrm{ads}\right)+e^{-}\to 2O^{-}\left(\mathrm{ads}\right)$$
(6)$$O^{-}\left(\mathrm{ads}\right)+e^{-}\to O^{2-}\left(\mathrm{ads}\right)$$

When the BiFeO_3_ sensor is in the reducing gas, just as isopropanol, oxygen species will react with isopropanol molecules and produce CO_2_ and H_2_O. In the reaction process, the trapped electrons will come back to the conduction band of BiFeO_3_ material, leading to an increase of electron concentration. However, bismuth ferrite is a p-type semiconductor, whose main carriers are holes. According to Equation (10), a small number of released electrons can neutralize the holes so that the resistance increases during the reaction with isopropanol. The above reactions can be described as follows [31]:(7)$$C_{3}H_{8}O\;\left(\mathrm{gas}\right)\to C_{3}H_{8}O\;\left(\mathrm{ads}\right)$$
(8)$$C_{3}H_{8}O\;\left(\mathrm{ads}\right)+9O^{-}\;\left(\mathrm{ads}\right)\to 3{\mathrm{CO}}_{2}\;\left(\mathrm{gas}\right)+4H_{2}O\;\left(\mathrm{gas}\right)+9e^{-}$$
(9)$$C_{3}H_{8}\;\left(\mathrm{ads}\right)+9O^{2-}\;\left(\mathrm{ads}\right)\to 3{\mathrm{CO}}_{2}\;\left(\mathrm{gas}\right)+4H_{2}O\;\left(\mathrm{gas}\right)+18e^{-}$$
(10)$$e^{-}+h^{.}\to \mathrm{Null}$$

When exposed to air again, the resistance value of BiFeO_3_ sensor will get back to its original value.

## 4. Conclusions

Pure BiFeO_3_ nanocrystals have been successfully fabricated by a simple wet chemical method. BiFeO_3_ nanocrystals sintered at 500 and 550 °C and the prepared gas sensors display almost the same performance.At the optimum working temperature of 240 °C, the fabricated sensor shows excellent isopropanol gas sensing properties with a high gas sensitivity of 31 exposed to 100 ppm isopropanol, fast response and recovery time (6 and 17 s), nice stability, and good selectivity to isopropanol.The sensor shows a perfect linear relationship between sensitivity and concentration in the range of 2–100 ppm at 240 °C and reaches the saturation point when the concentration is over 100 ppm. In addition, the measurement accuracy is of 1 ppm level. Therefore, the isopropanol gas sensor based on BiFeO_3_ nanocrystals can realize precise detection under 100 ppm concentration ranges and early warning over 100 ppm.The mechanism analysis reveals that the adsorbed oxygen ions may be mainly composed of O^2^.Conclusions above show that BiFeO_3_ nanocrystals are the superior candidate for a gas sensing application toward isopropanol detection.

## Figures and Tables

**Figure 1 materials-13-03829-f001:**
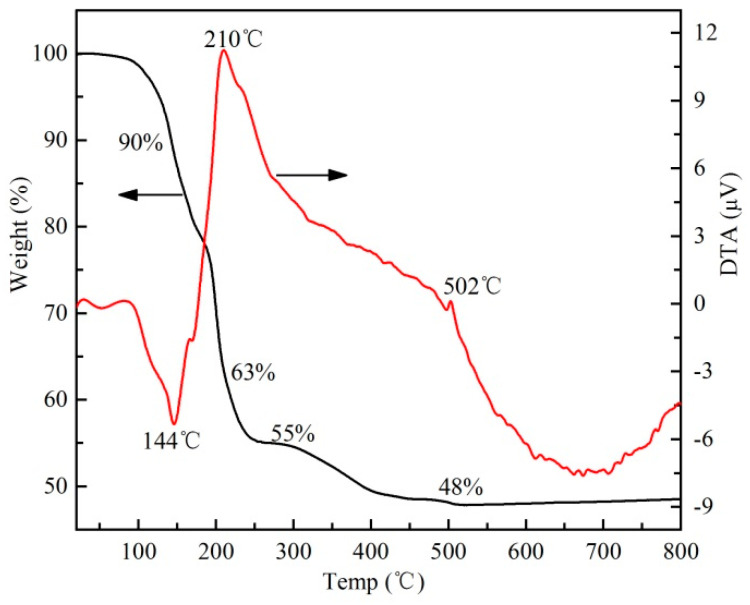
Thermogravimetric analysis and differential thermal analysis (TG/DTA) curve of BiFeO_3_ dry gel.

**Figure 2 materials-13-03829-f002:**
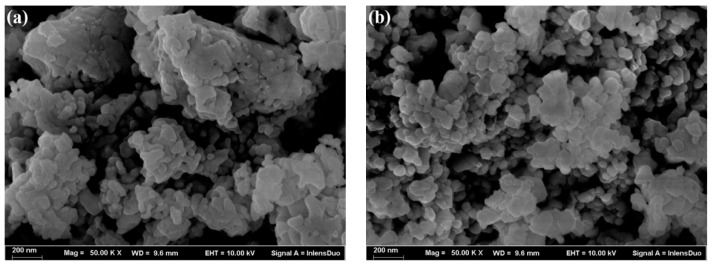
SEM images of BiFeO_3_ nanocrystals: (**a**) 500 °C and (**b**) 550 °C.

**Figure 3 materials-13-03829-f003:**
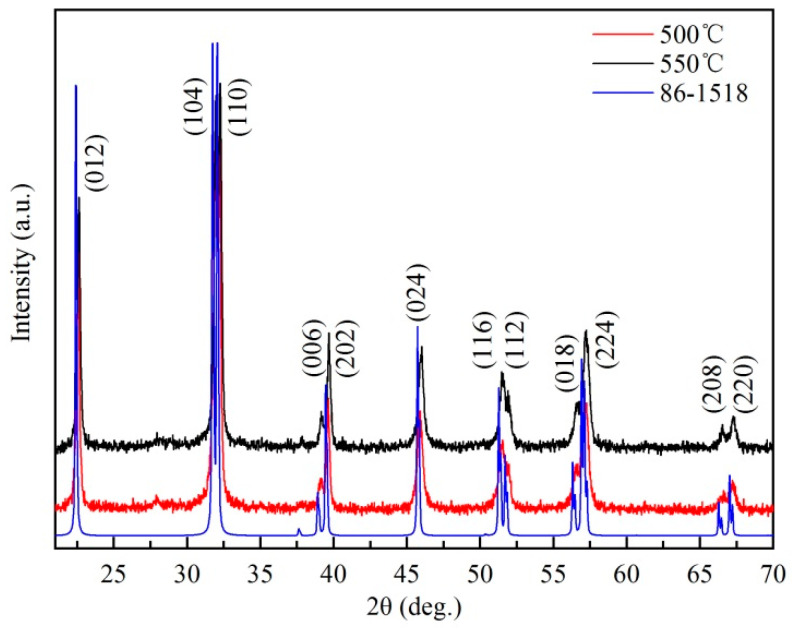
The XRD patterns of BiFeO_3_ nanocrystals.

**Figure 4 materials-13-03829-f004:**
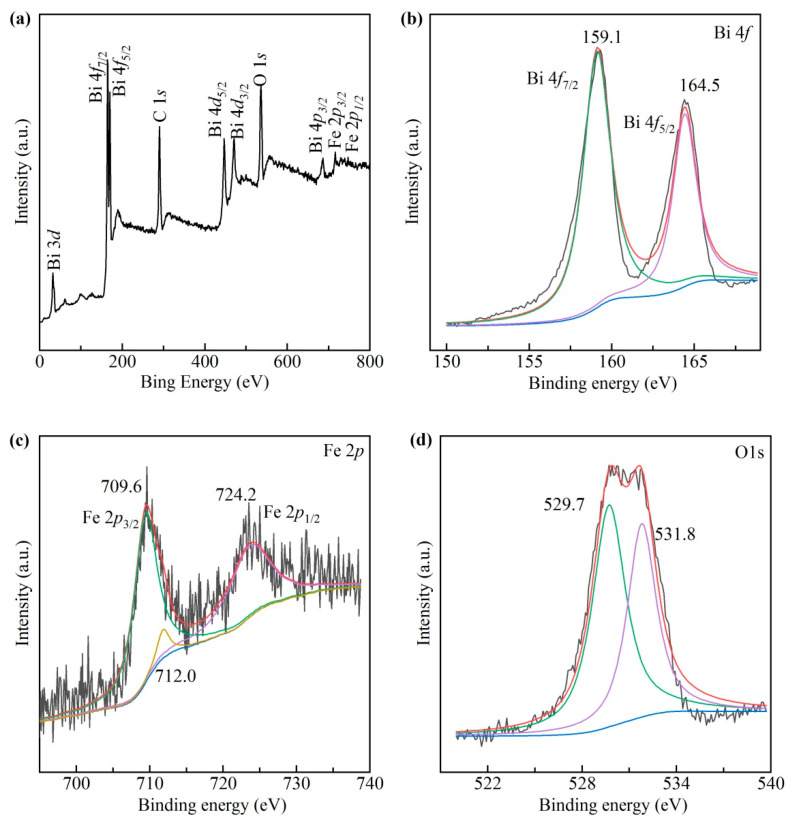
XPS survey spectra of BiFeO_3_ nanocrystals sintered at 500 °C: (**a**) Full XPS, (**b**) Bi 4f, (**c**) Fe 2p, and (**d**) O 1 s.

**Figure 5 materials-13-03829-f005:**
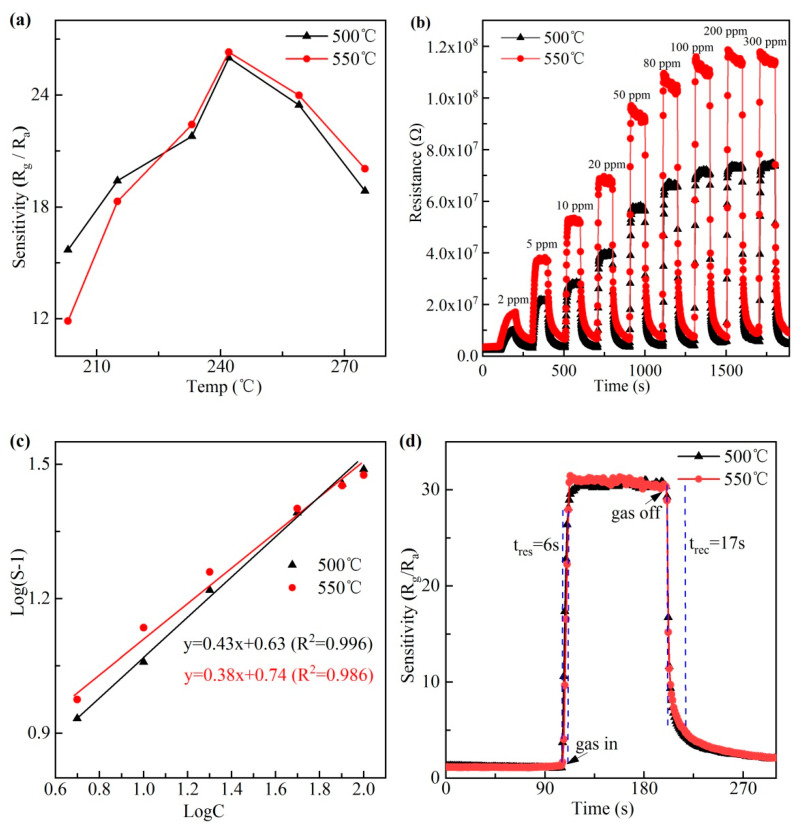
(**a**) Gas response of two BiFeO_3_ sensors to 50 ppm isopropanol at different operating temperatures, (**b**) dynamic resistance-time gas response of two sensors to isopropanol at 240 °C, (**c**) the corresponding log(S−1) versus logC curves of two sensors, and (**d**) response time and recovery time of two BiFeO_3_ sensors to 100 ppm isopropanol.

**Figure 6 materials-13-03829-f006:**
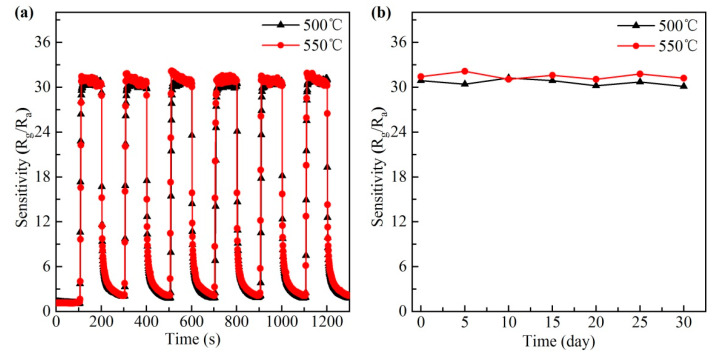
(**a**) Reversibility and (**b**) long-term stability of two BiFeO_3_ sensors to 100 ppm isopropanol at 240 °C.

**Figure 7 materials-13-03829-f007:**
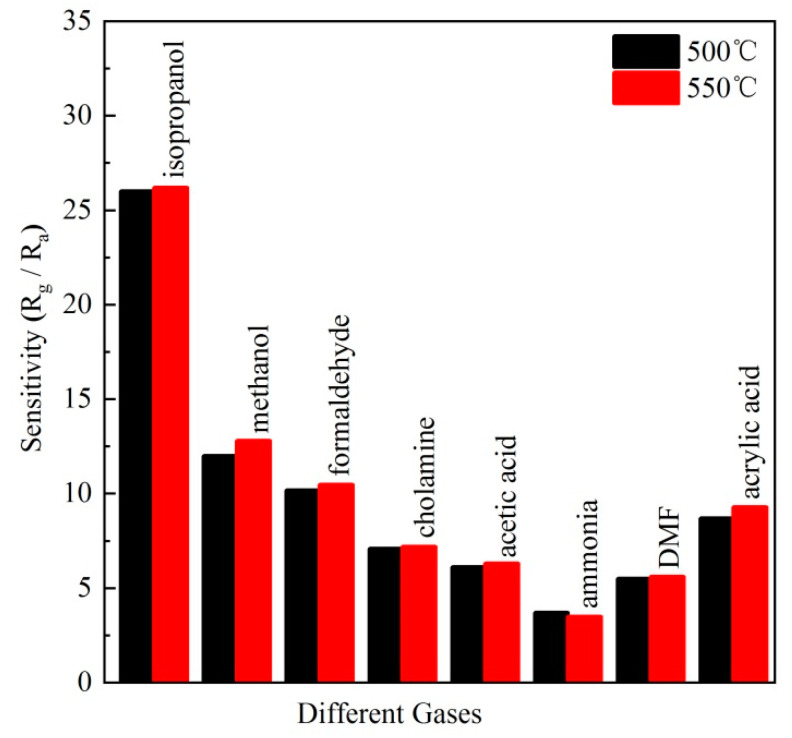
The selectivity of two BiFeO_3_ sensors to different gases (50 ppm) at 240 °C.

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
