# Peer review of "Fast Response Isopropanol Sensing Properties with Sintered BiFeO3 Nanocrystals"

_materials, 2020, doi:10.3390/ma13173829_

Round 1

Reviewer 1 Report

This manuscript reported the isopropanol sensing of BiFeO3 nanocrystals. This study can be publish after some major revision as follows:

1, I found out this report has high similarity with a few publics in literature as https://doi.org/10.1111/j.1551-2916.2009.03325.x, https://doi.org/10.1016/j.matlet.2017.03.091 and https://doi.org/10.1016/j.matlet.2019.02.129. The authors should discuss more in the introduction section to reveal the novelty of this study.

2, XRD peaks are obviously shifted to the right in order to compare with the JCPDS card, but the authors did not mention and explain?

3, In order to control the grain size and how it affect the sensing properties, the authors should provide some different sintering temperature.

4, how did the authors calculate the detection limit? Authors just indicated 1 ppm as the detection limit, but there is no more description.

5, Readers tend to see the change of resistance in exposure to the gas, so Figure 5b should plot the resistance curves and time dependence.

Author Response

Point 1: I found out this report has high similarity with a few publics in literature as

https://doi.org/10.1111/j.1551-2916.2009.03325.x, https://doi.org/10.1016/j.matlet.2017.03.091, https://doi.org/10.1016/j.matlet.2019.02.129.

The authors should discuss more in the introduction section to reveal the novelty of this study.

Response 1: Delete Line 54 of the original text”lacking the study on isopropanol”, add the relevant research result, propose the novelty of this study and modify as follows:

     Most of these reports indicate that BiFeO3 is very sensitive to alcohols and ketones and the gas sensitivities of BiFeO3 sensors are influenced by their morphology and particle size. Tong Tong [27] found that the sensitivities of BiFeO3 gas sensors to 50 ppm ethanol based on their disk-shaped particles (2μm) by hydrothermal method and nanoparticles (100nm) by co-precipition method are respectively 3.3 and 4 at 260℃. Yu Xuelian [29] controlled the size of nanoparticles by adjusting the growth time using sol-gel method to construct BiFeO3 tubular sensors and found that the smaller the particle size of BiFeO3 is, the greater the sensitivity is. The sensitivity of 30nm (100nm) particles to 50 ppm ethanol and acetone at 260°C are 40(27) and 36(24), respectively. It can be seen that the growth time and temperature need to be reduced in order to reduce the particle size and improve the sensitivity. In those reported BiFeO3 ethanol sensors, BiFeO3 nanoparticles have better gas sensing property than other nanostructures. It is probably because smaller nanoparticles have a larger specific surface area and abundant oxygen vacancy defects. So far, BiFeO3 gas sensitive material is limited to only a few organic gases and is still worth further exploring for many organic gases. Considering the gas sensing advantages of BiFeO3 nanoparticles and the toxicity of isopropanol, BiFeO3 nanocrystals are applied as the sensing materials to isopropanol in this paper.

Point 2: XRD peaks are obviously shifted to the right in order to compare with the JCPDS card, but the authors did not mention and explain?

Response 2: The deviation of the diffraction peak is explained as follows and modified accordingly:

     It is obvious that all the reflection peaks of BiFeO3 nanocrystal powders match with the spectrum of JCPDS Card No.86-1518, and all diffraction peaks of BiFeO3 nanocrystals are obviously shifted to the right by about 0.13º, suggesting that BiFeO3 nanocrystal powders are of the rhombohedral distorted perovskite structure (Space group R3c). It is probably because there are a large number of oxygen vacancies in the body and on the surface of BiFeO3 nanocrystals, greatly increasing the lattice distortion and leading to the diffraction peak shift to the right. The diffraction peaks of two sintered powders did not deviate significantly and the consistency of their peak heights and peak widths once again indicates the comparability of the two nanocrystals.

Point 3: In order to control the grain size and how it affect the sensing properties, the authors should provide some different sintering temperature.

Response 3:  According to the report [27, 29], the grain size can greatly affect gas sensing properties, the lower the growth temperature is, the smaller the particle size is, and the higher the gas sensitivity is. Therefore, in order to achieve experiment purposes of small particle size and high sensitivity, two sintering temperatures near 500°C is selected as the growth temperatures of BiFeO3 nanocrystals. Modify as follow:

Therefore, to control the grain size, different grown temperatures (500 and 550 °C) are adopted as the sintering temperature of BiFeO3 nanocrystals.

Point 4: how did the authors calculate the detection limit? Authors just indicated 1 ppm as the detection limit, but there is no more description.

Reponse 4:  In this paper, the detection limit of BiFeO3 to isopropanol is estimated by the concentration-sensitivity curve(Figure 5b). Figure 5b shows that the sensitivity of BiFeO3 sensor to 2 ppm isopropanol is 4.8 when liquid organic gas was injected by microinjector within the experimental error range. In general, the sensitivity is linear to the low concentration gas. So the sensitivity of BiFeO3 sensor to 1ppm isopropanol is about 2.4, which is easy to be observed. Therefore, under our test conditions, the low detection limit of BiFeO3 sensor to isopropanol reaches 1ppm level.

Point 5: Readers tend to see the change of resistance in exposure to the gas, so Figure 5b should plot the resistance curves and time dependence.

Response 5: According to the requirements of review, the sensitivity –time curve is modified into the resistance-time curve.  The initial resistances of BiFeO3 sensors grown at 500 and 550°C in air are 2.2515×106 and 3.5876×106 Ω, respectively. The sensitivity-time curve is automatically obtained by the gas sensitive testing software according to the formula S=Rg/Ra.

The sensitivity of two sensors is almost the same after calculation by the testing software.

Reviewer 2 Report

  1. Abstract can be reformulated without repeating some groups of words, as suggested:In view of the toxicity of isopropanol gas and the potential excellent gas-sensitive  properties of BiFeO3 nanocrystals, the isopropanol gas sensing performance of BiFeO3-nanocrystal were studied.
  2. Llines 30 31-Please unify the two sentences: Therefore, precise detection and early warning of isopropanol are of great importance and gas sensors are considered to be an effective approach.
  3. Line 39 I suggest instead of" inferior points", weaknesses and instead of "have been made into all kinds of nanostructures"  several types of architectures have been designed
  4. Lines 48-50: please unify the two sentences.
  5. Line 54 "lacking the study on isopropanol "is not appropriate because there are many reported detection studies that included isopropanol:

    Sensors and Actuators B: Chemical

    Volume 237, December 2016, Pages 776-786;

    Analytica Chimica Acta

    Volume 757, 13 December 2012, Pages 75-82   and many more, that have to be correctly cited.
  6. Section 2.1 is not appropriately described. The method lacks to present the amounts of reagents, their concentration (for instance dilute nitric acid ?)
  7. Section 2.3 is also  not appropriately described. What are the used quantities?
  8. Section 3.1 Please assign to what is due "When the temperature reaches 300 ℃, the weight loss ratio is 45%".

Author Response

Point 1: Abstract can be reformulated without repeating some groups of words, as suggested: In view of the toxicity of isopropanol gas and the potential excellent gas-sensitive properties of BiFeO3 nanocrystals, the isopropanol gas sensing performance of BiFeO3-nanocrystal were studied.

Response 1: BiFeO3 nanocrystals were applied as the sensing material to isopropanol.

Point 2: Lines 30 31-Please unify the two sentences: Therefore, precise detection and early warning of isopropanol are of great importance and gas sensors are considered to be an effective approach.

Response 2: Therefore, as an effective approach, using gas sensors to realize precise detection and early warning of isopropanol is of great importance.

Point 3: Line 39 I suggest instead of “inferior points”, weaknesses and instead of “have been made into all kinds of nanostructures” Several types of architectures have been designed.

Response 3: To improve these weaknesses, several types of architectures such as mesoporous ball-flower structures [11], hollow nanospheres [12] have been designed.

Point 4: Lines 48-50: please unify the two sentences.

Reponse 4: Given the oxides' commonalities, lots of oxygen vacancies are in the body and the surface of BiFeO3, while the surface oxygen vacancies provide reliable locations for external oxygen molecules.

Point 5: Line 54”lacking the study on isopropanol” is not appropriate because there are many reported detection studies that included isopropanol: Sensors and Actuators B: Chemical Volume 237, December 2016, Page 776-786;  Analytica Chimica Acta Volume 757, 13 December 2012, Pages 75-82 and many more  that have to be correctly cited

Response 5:  Delete Line 54 of the original text”lacking the study on isopropanol”, add the relevant research result, propose the novelty of this study and modify as follows:

 Most of these reports indicate that BiFeO3 is very sensitive to alcohols and ketones and the gas sensitivities of BiFeO3 sensors are influenced by their morphology and particle size. Tong Tong [27] found that the sensitivities of BiFeO3 gas sensors to 50 ppm ethanol based on their disk-shaped particles (2μm) by hydrothermal method and nanoparticles (100nm) by co-precipition method are respectively 3.3 and 4 at 260°C. Yu Xuelian [29] controlled the size of nanoparticles by adjusting the growth time using sol-gel method to construct BiFeO3 tubular sensors and found that the smaller the particle size of BiFeO3 is, the greater the sensitivity is. The sensitivity of 30nm (100nm) particles to 50 ppm ethanol and acetone at 260°C are 40(27) and 36(24), respectively. It can be seen that the growth time and temperature need to be reduced in order to reduce the particle size and improve the sensitivity. In those reported BiFeO3 ethanol sensors, BiFeO3 nanoparticles have better gas sensing property than other nanostructures. It is probably because smaller nanoparticles have a larger specific surface area and abundant oxygen vacancy defects. So far, BiFeO3 gas sensitive material is limited to only a few organic gases and is still worth further exploring for many organic gases. Considering the gas sensing advantages of BiFeO3 nanoparticles and the toxicity of isopropanol, BiFeO3 nanocrystals are applied as the sensing materials to isopropanol in this paper.

Point 6: Sections 2.1 is not appropriately described. The method lacks to present the amounts of reagents, their concentration (for instance dilute nitric acid?)

Response 6: First, 0.006 mol Bi(NO3)3·5H2O (2.938g)  was added into the dilute nitric acid  (2ml concentrated nitric acid whose content is 65-68%  was mixed  by 8ml water) and stirred for half an hour to obtain a colorless transparent solution (Bi-based solution). Next, 0.006 mol Fe(NO3)3·9H2O (2.461g) was dissolved in deionized water (10ml)  to a form a transparent red solution (Fe-based solution).

Point 7: Sections 2.3 is also not appropriately described. What are the used quanlities?

Response 7: 200 mg BiFeO3 nanocrystals were mixed with deionized water (0.2 ml) by grinding in an agate mortar to form a paste.  

Point 8: Section 3.1 Please assign to what is due “When the temperature reaches 300 °C, the weight loss ratio is 45%”.

Response 8: When the temperature reaches 300°C, the weight loss ratio of the dry gel is 45%.

Round 2

Reviewer 1 Report

The authors have addressed all the comments. After carefully evaluation, I suggest to accept it in the present form.

Reviewer 2 Report

The revised manuscript entitled:

Fast response isopropanol sensing properties with sintered BiFeO3 nanocrystals

was highly improved, so that I agree it can be published.